# CRISPR/Cas9 Mediated Knockout of Cyclooxygenase-2 Gene Inhibits Invasiveness in A2058 Melanoma Cells

**DOI:** 10.3390/cells11040749

**Published:** 2022-02-21

**Authors:** Cathleen Haase-Kohn, Markus Laube, Cornelius K. Donat, Birgit Belter, Jens Pietzsch

**Affiliations:** 1Helmholtz-Zentrum Dresden-Rossendorf (HZDR), Department of Radiopharmaceutical and Chemical Biology, Institute of Radiopharmaceutical Cancer Research, 01328 Dresden, Germany; m.laube@hzdr.de (M.L.); b.belter@hzdr.de (B.B.); j.pietzsch@hzdr.de (J.P.); 2Helmholtz-Zentrum Dresden-Rossendorf (HZDR), Translational TME-Ligands, Institute of Radiopharmaceutical Cancer Research, 01328 Dresden, Germany; c.donat@hzdr.de; 3Faculty of Chemistry and Food Chemistry, School of Science, Technische Universität Dresden, 01069 Dresden, Germany

**Keywords:** chemosensitivity, CRISPR/Cas9-knockout, 3D-tumor spheroid models, malignant melanoma, radiosensitivity, selective COX-2 inhibitors, tumor hypoxia

## Abstract

The inducible isoenzyme cyclooxygenase-2 (COX-2) is an important hub in cellular signaling, which contributes to tumor progression by modulating and enhancing a pro-inflammatory tumor microenvironment, tumor growth, apoptosis resistance, angiogenesis and metastasis. In order to understand the role of COX-2 expression in melanoma, we investigated the functional knockout effect of COX-2 in A2058 human melanoma cells. COX-2 knockout was validated by Western blot and flow cytometry analysis. When comparing COX-2 knockout cells to controls, we observed significantly reduced invasion, colony and spheroid formation potential in cell monolayers and three-dimensional models in vitro, and significantly reduced tumor development in xenograft mouse models in vivo. Moreover, COX-2 knockout alters the metabolic activity of cells under normoxia and experimental hypoxia as demonstrated by using the radiotracers [^18^F]FDG and [^18^F]FMISO. Finally, a pilot protein array analysis in COX-2 knockout cells verified significantly altered downstream signaling pathways that can be linked to cellular and molecular mechanisms of cancer metastasis closely related to the enzyme. Given the complexity of the signaling pathways and the multifaceted role of COX-2, targeted suppression of COX-2 in melanoma cells, in combination with modulation of related signaling pathways, appears to be a promising therapeutic approach.

## 1. Introduction

Cyclooxygenase-2 (COX-2; *PTGS2* (prostaglandin-endoperoxide synthase-2) is an inducible enzyme that converts arachidonic acid to prostaglandins. The upregulation of COX-2 by a variety of factors, including cytokines and growth factors, and the resulting increase in prostaglandin E2 (PGE2) levels may promote chronic inflammatory conditions, thus initiating tumor development [1,2,3]. COX-2 overexpression is also induced by tumor microenvironmental (TME) factors, such as tumor hypoxia [4], and has manifold pathophysiological consequences [5,6]. Via multiple signals, COX-2 contributes to tumor progression [7,8], for instance, upregulation of the vascular endothelial growth factor (VEGF) inducing apoptosis resistance and angiogenesis [9]. Moreover, enhanced tumor growth through upregulation of the epidermal growth factor receptor (EGFR) and cancer cell survival through interaction with phosphatidylinositol 3-kinase/(PI3K)/AKT [9,10] also trigger tumor progression.

There is a great need for new and improved treatment strategies in melanoma and metastatic melanoma patients. Modulation of COX-2 might offer this, as the enzyme frequently is expressed in malignant melanoma and correlates significantly with poor survival in patients [11,12,13]. High levels of COX-2 have been detected in both murine and human melanoma models [11,14,15]. Here, different COX-2 pathway and transcription factors are important for their role in inflammation and melanoma cancer progression. In particular, the most studied and commonly altered pathways in melanoma are the mitogen-activated protein kinase (MAPK), PI3K [16,17], and Janus kinase-2/signal transducer and activator of transcription 3 (JAK-2/STAT3) [8]. COX-2 inhibition can be achieved by either non-selective nonsteroidal anti-inflammatory drugs (NSAIDs) or selective COX-2 inhibitors (COXIBs) and has anti-inflammatory, antipyretic and analgesic effects. Furthermore, potential anti-tumor effects and radiosensitizing actions of COX-2 inhibitors have been intensively investigated [18,19,20]. In contrast to the long-term treatment of chronic inflammatory diseases, short-term treatments in combination with other approaches, such as the adjuvant use of COX-inhibitors in chemo- or radiotherapy, are being discussed for tumor therapeutic approaches. However, an adjuvant use of selective COX-inhibitors in the very promising immunotherapies should be critically questioned.

The question remains as to whether the effects of selective COX-2 inhibitors occur in a COX-2 expression-dependent or independent manner in tumor cells [21,22,23,24,25,26]. For example, many reports indicate that the COX-2 selective inhibitor celecoxib does not require the presence of COX-2 to implement its anti-tumor effects [22,25,27].

A closer examination of cancer cell regulation and signaling pathways of COX-2 leads to a better understanding of the complexity of COX-2-mediated tumorigenesis. In this regard, in vitro and in vivo models in which COX-2 expression is knocked down can provide important insights into the role of COX-2-related molecular and cellular pathways for specific disease models, and to characterize the pharmacological profile of inhibitors in terms of their COX-2-dependent and independent effects. Until now, techniques such as siRNA, shRNA and microRNA-based gene silencing were utilized to generate COX-2 knockdown models in glioblastoma, breast cancer, bladder tumors and melanoma (Figure 1) [10,12,28,29,30,31,32]. Since the discovery of the CRISPR/Cas9 technique, it has been widely and successfully applied in biomedical research, modifying gene activation and repression, and leading to improved animal models for the development of new therapeutic approaches [33,34,35]. The use of the novel CRISPR/Cas9 technique by Ercolano et al., further allowed the generation of a COX-2-knockout model in B16-F10 melanoma cells unraveling marked effects in vitro and in vivo [29]. The aim of this work was to study cellular and molecular mechanism in human A2058 cells and CRISPR/Cas9-mediated COX-2 knockout cells to understand the enzyme’s role in the development and progression of melanoma. Here, subcutaneous tumor growth in vivo as well as migration, invasion and colony formation, growth rate and metabolic parameters (hypoxia) were investigated. Furthermore, sensitivity to treatment with COXIBs and/or irradiation as well as the influence on signaling pathways in monolayer cultures and/or multicellular tumor spheroids was measured in vitro.

## 2. Materials and Methods

### 2.1. Cell Culture, Spheroid Cell Culture and Culture Conditions

Human melanoma cell line A2058 (No. CRL-11147) was purchased from ATCC and cultured in Dulbecco’s modified Eagle medium (DMEM) supplemented with 10% (*v*/*v*) fetal calf serum and 1 U/mL penicillin/streptomycin (all reagents from Biochrom, Berlin, Germany) under normoxic conditions. For hypoxia experiments, cells were cultured under reduced oxygen level as described previously [36]. For the generation of tumor spheroids, cells were cultured in complete growth medium supplemented with 0.24% (*w*/*v*) methylcellulose as described [36]. Tumor spheroids were maintained for 5, 12 or 19 days and used for the corresponding experiments.

### 2.2. Knockdown of COX-2 with CRISPR/Cas9

A CRISPR/Cas9 knockdown kit against human COX-2 was purchased from Santa Cruz Biotechnology (CA, Heidelberg, Germany). Transfection was performed as recommended by the manufacturer. Briefly, 2 × 10^5^ A2058 cells were seeded into six-well plates 24 h prior to transfection. UltraCruz^®^ Transfection Reagent (Santa Cruz Biotechnology, Heidelberg, Germany) was used at a final concentration of 1% together with a total of 2 µg plasmid (1 µg Cox-2 CRISPR/Cas9 KO Plasmid with 1 µg COX-2 HDR Plasmid) per well. The Plasmid DNA/UltraCruz^®^ complexes were made up in serum-free growth medium. Cells were maintained for 24 h before returning to growth medium and puromycin selection (0.5 µg/mL) (Santa Cruz Biotechnology, Heidelberg, Germany). The COX-2 HDR Plasmid is recommended for the co-transfection with COX-2 CRISPR/Cas9 KO plasmid and designed for repair of the site-specific Cas9-induced DNA cleavage within the PTGS2 gene. During repair, the Cox-2 HDR plasmid incorporates a puromycin resistance gene to enable selection of stable knockout (KO) cells and an RFP gene to visually confirm transfection. RFP-positive and puromycin-resistant cells were selected by single-cell collection (monoclonal population) and named A2058-COX-2KO.

### 2.3. PGE_2_ Measurements

PGE_2_ levels in cultured supernatants form A2058 and three different clones of A2058-COX-2KO were evaluated by using commercially available enzyme-linked immunosorbent assay kit (ENZO Life Science, Lörrach, Germany, ADI-930-001) according to manufacturer´s recommendations.

### 2.4. Growth Rate Analysis

For growth assays, cells were plated with 5 × 10^4^ cells per well in 6-well plates. At 1, 2, 5, 6, 7 and 9 days cells were trypsinized and counted using a CASY1 cell counter (Model TT, Schaerfe System, Reutlingen, Germany). Three independent experiments were performed, each in triplicate. For each cell line the diameter of 12 spheroids was measured at day 2, 5, 7, 9, 12, 14, 16, 19 and 21 using an inverted microscope (AxioVert 40CFL, Carl Zeiss MicroImaging, Jena, Germany) and the software package AxioVision SE64.

### 2.5. Scratch Wound Cell Migration and Invasion Assay Using the IncuCyte^®^ Live-Cell Monitoring System

For migration, 6 × 10^4^ cells were seeded in a 96-well ImageLock plate (Essen BioScience, Sartorius, Royston, UK) in a total of 100 µL, and allowed to adhere overnight. The scratches in 96-well format were applied using the IncuCyte^®^ WoundMaker tool (part of the IncuCyte^®^ Live-Cell Monitoring System, Essen BioScience, Sartorius, Royston, UK). It is a pin-based tool that applies one scratch per well in all wells in parallel. After the scratch, cells were washed twice, and then incubated either with 10% serum or without (0%) fetal calf serum (FCS) to investigate the influence of proliferation. For the invasion assay, the 96-well ImageLock plate was coated with Matrigel, cells were seeded and incubated overnight. On the next day, media were replaced and overlay cells were covered with matrigel on top layer. After 30 min incubation, scratches are applied using the IncuCyte^®^ WoundMaker tool, and media with (10%) or without (0%) FCS was added. The plates were placed in the IncuCyte^®^ for imaging every 12 h for 6 to 7 days. The IncuCyte^®^ Scratch Wound Cell Migration and Invasion Software Analysis Module allows automated detection and quantification of wound properties using a 10× objective.

### 2.6. Clonogenic Assay

Clonal growth assay was used to compare the clonogenicity of A2058 vs. A2058-COX-2KO. Furthermore, the sensitivity of the melanoma cell lines toward irradiation and in combination with celecoxib and rofecoxib at 1 and 10 µM, respectively, was analyzed. Cells were plated into 6-well plates as follows: 2000 cells/well for sham and 2 Gy X-ray; 3000 cells/well for 4 Gy; 5000 cells/well for 6 Gy and 10,000 cells/well for 10 Gy X-ray. Experiments were performed in duplicate per data point. The cells were then permitted to attach for 24 h at 37 °C. A stock solution of celecoxib and rofecoxib was prepared in DMSO (Sigma-Aldrich, Taufkirchen, Germany) and stored at −20 °C. Before the beginning of the experiment, the stock solution was diluted to the appropriate concentration (1 µM or 10 µM), and then cells were incubated for 1 h at 37 °C either with celecoxib, rofecoxib or 1% DMSO as sham. Cells were then immediately irradiated using a Maxishot system (YXLON International, Hamburg, Germany; 200 kV, filtered with 0.5 mm Copper). The absorbed dose was measured using a UNIDOS dosimeter (PTW, Freiburg, Germany). The dose-rate was approximately 1.1 Gy/min at 20 mA. The cells were further maintained at 37 °C for 7 days to allow formation of colonies and then stained with 0.5% crystal violet (Sigma) in absolute methanol. Colonies greater than 50 cells were counted visually under an inverted microscope (2000C, Carl Zeiss AG, Jena, Germany). Plating efficiency was calculated as the ratio of the number of colonies to the number of seeded cells. Relative plating efficiency was calculated as the ratio of the plating efficiency of treated cells to the plating efficiency of sham-control × 100%. Three independent experiments, each in duplicate, were performed.

### 2.7. Cell Viability Assay

The cytotoxicity of celecoxib and rofecoxib was determined by MTT assay. Cell viability assay was performed as previously described [37]. After cells being cultured for 24 h, the medium was replaced with new DMEM medium containing DMSO (control) or various concentrations of celecoxib or rofecoxib (12.5, 25, and 50 µM) for 24 and 48 h.

### 2.8. [^18^F]FDG and [^18^F]FMISO Radiotracer Uptake

To characterize glucose consumption, normoxia and intrinsic/extrinsic hypoxia in melanoma cell monolayers and tumor spheroids, uptake experiments with 2-[^18^F]fluoro-2-deoxy-D-glucose ([^18^F]FDG) and [^18^F]fluoromisonidazole ([^18^F]FMISO; specific activity at application time, 115 GBq/µmol) were performed. Therefore, monolayer cells were subjected under normoxic and to extrinsic hypoxic conditions for 24, 48 and 72 h. At the indicated time point, the growth medium was removed and 0.5 MBq [^18^F]FDG or [^18^F]FMISO in 500 µL DMEM was added, followed by incubation either under normoxic or under hypoxic conditions for another 4 h. Subsequently, the medium was removed. The cells were washed three times with ice-cold PBS, lyzed with 500 µL 0.1 M NaOH/SDS, transferred into small tubes and measured using a Wizard gamma counter (Canberra-Packard, Meriden, CT, USA). Cellular uptake of the [^18^F]FDG or [^18^F]FMISO was determined at least eight times in three independent experiments. For the analysis of intrinsic hypoxia spheroids after 4, 11 and 18 days of cultivation were used. The growth medium was discarded and 0.15 MBq [^18^F]FDG or [^18^F]FMISO in 100 mL DMEM was added for 4 h. After incubation, tumor spheroids were transferred using a vacuum pump to a 96-well membrane filter plate. The filter membranes were washed three times with ice-cold PBS, transferred into small tubes, and measured. Cellular uptake of [^18^F]FDG or [^18^F]FMISO was determined at least six times in three independent experiments.

### 2.9. Western Blot Analysis

SDS-PAGE and Western blot was performed as described previously [38]. PVDF membranes were incubated with anti-COX-2 (ab15191; 1:500, Abcam, Berlin, Germany) and anti-β-actin (A5060; 1:1000, Sigma–Aldrich, Taufkirchen, Germany) antibodies and then probed with secondary antibodies, conjugated to horseradish peroxidase (Sigma–Aldrich, Taufkirchen, Germany). Immunoblots were detected by enhanced chemiluminescence with a Bio-Imaging-System MF-ChemBIS3.2 (Biostep, Burkhardtsdorf, Germany). Each experiment was repeated at least three times.

### 2.10. Animals and Generation of A2058 and A2058-COX-2KO Melanoma Xenografts

All animal experiments were carried out according to the guidelines of the German Regulations for Animal Welfare. The protocols were approved by the local Ethical Committee for Animal Experiments (reference number DD24.1-5131/449/49). Generation of tumor xenografts was performed as described elsewhere [39]. In brief, female NMRI-nude mice (Rj:NMRI-Foxn1nu, Janvier Labs, Le Genest-Saint-Isle, France) were subcutaneously injected with 4 × 10^6^ A2058 or A2058-COX-2KO cells, each in 100 µL 0.9% *v*/*v* NaCl, into the right hind leg. Tumor size was monitored three times a week by caliper measurements and tumor volume was calculated using the formula V = π/6 × (tumor length × tumor width^2^). Tumor-bearing mice were included in the experiments when tumors reached a volume of at least 400 to 1500 mm^3^. Mice were sacrificed using CO_2_ inhalation and cervical dislocation.

### 2.11. Optical In Vivo Imaging

Optical tumor imaging in mice was performed on a preclinical In-Vivo Xtreme imaging system (Bruker, Billerica, MA, USA) under general anesthesia with inhalation of 10% (*v*/*v*) desflurane (Baxter, Unterschleißheim, Germany) in 30% (*v*/*v*) oxygen/air. Location and signal intensity of RFP A2058-COX-2KO xenografts were assessed using bioluminescence imaging at λEx/Em = 550/600 nm and non-specific fluorescence was captured at λEx/Em = 480/535 nm. Tumor uptake was analyzed in processed images, showing specific fluorescence/non-specific fluorescence ratios. X-ray images were merged with bioluminescence images.

### 2.12. Protein Phosphorylation Assay

In order to identify potential downstream signaling targets of COX-2, we compared the two cell lines A2058 and A2058-COX-2KO in a pilot experiment using a “Cancer Phospho Signaling Antibody Array” (269 targets; PCS248; tebu-bio, Le Perray-en-Yvelines, France). The antibodies on the array are covalently immobilized on a high quality glass surface coated with proprietary 3-D polymer materials, which were designed to promote high binding efficiency and specificity. All arrays were printed on standard-size microscope slides, and each slide contained one complete array with six replicates for each antibody. For normalization, within each array slide, the median value of the average signal intensity for all antibodies on the array was determined. This value was presented as a median signal. Using the normalized data, the fold change between A2058 and A2058-COX-2KO was determined and the results were highlighted as red (increase in expression, ratio ≥ 2) or green (decrease in expression, ratio ≤ 0.5). The antibody array procedure was performed as recommended by the manufacturer. The array image quantification and analyses was performed at tebu-bio (Le Perray-en-Yvelines, France).

### 2.13. Statistical Analysis

Descriptive data were expressed as means ± SD or means ± SEM. The number of n in the figure legends represents the number of independent experiments. Statistical analyses were carried out by one-way analysis of variance with post-hoc Bonferroni’s method using OriginPro 2018G (Additive GmbH, Friedrichsdorf, Germany). *p* values less than 0.05 were considered to be statistically significant.

## 3. Results

### 3.1. COX-2 Knockout Slowes Subcutaneous Tumor Growth

CRISPR/Cas9 technology was utilized to generate a stable knockout of COX-2 expression in human melanoma A2058 cells and to evaluate its results in vitro and in vivo. The COX-2 knockout (COX-2KO) in the generated cell line was confirmed using RFP-FACS analysis and Western blotting (Figure 2a and Appendix A). To test whether the COX-2 knockout has an impact for the PGE_2_ release, we performed a prostaglandin E_2_ ELISA. The CRISPR/Cas9 mediated knockout of COX-2 completely inhibits the PGE_2_ production compared to the A2058 control cells (Appendix A). To prove that the generated COX-2KO was stably incorporated into the cells and alteration had an impact on in vivo behavior of the resulting tumor, human melanoma A2058 and A2058-COX-2KO cells were subcutaneously implanted into NMRI-nude mice. The RFP-Fluorescence of the COX-2 knockout cells allowed imaging of the subcutaneous tumors, as shown in Figure 2b, over the whole investigation period showing that the knockout was maintained in vivo. Furthermore, a delay in the tumor growth and hence an impact on the in vivo tumor model was observed for A2058-COX-2KO compared to A2058 (Figure 2c). The average time to reach a tumor volume of 250 mm^3^ was about 15 days for A2058 control cells and 27 days in mice implanted with A2058-COX-2KO cells (Figure 2c). In animals injected with A2058 wild-type cells, tumors grew in 14 of 14 animals, and in animals injected with A2058-COX-2KO, only 11 of the 14 animals developed tumors. We further confirmed these result by subcanteous injection of two additional clones of A2058-COX-2KO cell in NMRI-nude mice (Appendix A).

### 3.2. Cox-2 Knockout Changes the Metabolic Activity under Normoxia and under Conditions of Experimental Hypoxia

Cell viability of A2058 and A2058-COX-2KO was determined by growth rate analysis (Figure 3a–c). In contrast to the in vivo results, no difference was observed between both cell lines (Figure 3a). We further confirmed these in vitro results with two additional clones of A2058-COX-2KO cell (Appendix A). The disagreement between the observed in vitro and in vivo results is most likely caused by the surrounding microenvironment. Under standard in vitro conditions, it can be assumed that cells of both lines are surrounded by optimal nutrient availability, resulting in a similar growth rate. In vivo conditions present a tumor microenvironment, where insufficient nutrients or nutrient gradients exists resulting in an advantage for the wild-type cells. Metabolic changes were characterized by studying the uptake of the radiotracers [^18^F]FDG and [^18^F]FMISO. [^18^F]FDG was used to determine glucose uptake in these cells under normoxic conditions. Initial glucose uptake up to 60 min post addition was not different between WT and knockout cells. However, uptake in A2058 WT cells was significantly increased at 120 min over the knockout cells, indicating lower metabolic activity in A2058-COX-2KO cells (Figure 3b). Next, extrinsic hypoxia in the adherent cell monolayer cultures by oxygen-restriction was investigated using a hypoxia imaging probe [^18^F]FMISO, a radiotracer that accumulates in hypoxic cells via covalent binding to macromolecules after reduction of its nitro group. As shown in Figure 3c, radiotracer uptake under normoxic conditions displayed no differences between A2058 and A2058-COX-2KO. In contrast, the cellular uptake of [^18^F]FMISO at 24 h under (extrinsic) hypoxic conditions in A2058 significantly increased at 240 min when compared to normoxic controls. In A2058-COX-2KO, [^18^F]FMISO uptake did not significantly increase in hypoxic cells compared to normoxic controls or A2058-hypoxia cells.

### 3.3. COX-2 Knockout Reduces Cell Invasion and Motility

Next, we investigated whether the knockout of COX-2 affected the motility and invasive properties in these cells. Further, we treated the cells with and without fetal calf serum (FCS) to investigate the influence of proliferation. We first examined the effect on the migration of A2058 cells in a scratch wound healing assay. Results showed that A2058-COX-2KO displayed lower levels of migration compared to A2058 cells during the first three days in the presence of 10% FCS (Figure 4a, corresponding images in Appendix A). Over a long time-period no differences were observed. Migration in the absence of FCS revealed slow migration in both cell lines up to day 5. Over the observed time period, it seems that A2058-COX-2KO stop migration due to the inhibitory proliferation effects. Next, we surrounded the cells with Matrigel mimicking the extracellular matrix to evaluate the invasion capacity. As shown in Figure 4b, A2058-COX-2KO cells exhibited a lower rate of invasion compared to A2058, both in media with and without FCS (corresponding images in Appendix A).

### 3.4. Cox-2 Knockout Alters the Tumor Spheroid Formation Capability and Changes the Metabolic Activity and Intrinsic Hypoxia

To better reflect the physiological situation of hypoxia in avascular tumors or micrometastases, tumor spheroids were generated. For both cell lines, an optimized cell number of 2000 cells per well was used for spheroid generation, allowing an exponential growth pattern over 19 days. The growth curves of A2058 and A2058-COX-2KO spheroids show the diameter of single spheroid over a time of 21 days (Figure 5a). A2058-COX-2KO spheroids grew faster than A2058 spheroids until day 14. At day 19, both cell lines reached a similar size with a critical diameter of around 1000 µm. Then, the spheroids retained their size for 2 days before they disaggregated. The spheroids of both cell lines differed in their morphology. In brief, A2058 cells showed spontaneous formation of a single rounded cell aggregate at day 5 (Figure 5b). Compared to A2058-COX-2KO, A2058 spheroids were smaller and more compact, forming solid spheroids with a smooth surface and round shape (day 12 and 19). In contrast, A2058-COX-2KO aggregated into loose clusters confirmed by HE staining in Figure 5b, and had a non-spherical shape, which were more irregular and surrounded by a sphere of dead/non-aggregated cells overtime. While the diameter was increased, the total number of cells is not significantly different from the A2058 wild-type cells.

To characterize intrinsic hypoxia conditions in A2058 and A2058-COX-2KO spheroids, we investigated the histological distribution of pimonidazole. HE staining was performed in order to analyze the internal structure (Figure 5b). The positive immunohistochemical staining of pimonidazole confirmed the formation of hypoxic areas in 12-day and 19-day spheroids for both cell lines, respectively. In the 12-day spheroids, limited degrees of necrosis were visible. Up to day 19, a necrotic core had formed, with a bigger size for the 19-day-A2058COX-2KO.

Radiotracers [^18^F]FDG and [^18^F]FMISO were used to investigate metabolic differences in the spheroids. A2058 spheroids showed a time-dependent increase in [^18^F]FDG uptake until day 12, and a good agreement between uptake of [^18^F]FDG, spheroid diameter and protein content was observed. [^18^F]FDG uptake decreased in A2058 spheroids at day 19, likely caused by necrosis in the growing spheroids (Figure 5c). A2058-COX-2KO spheroids did not display an agreement between [^18^F]FDG uptake and spheroid diameter. Uptake of [^18^F]FDG was found to be lower over time and significantly decreased at day 19, when spheroid growth reached the maximum in A2058-COX-2KO spheroids (Figure 5d).

For the characterization of intrinsic hypoxia in growing spheroids, [^18^F]FMISO uptake was measured. Spheroids of A2058 (Figure 5e) revealed a significant increase in [^18^F]FMISO uptake in 12-day-A2058 and 19-day-A2058 compared with 4-day-A2058 spheroids. In A2058-COX-2KO (Figure 5f), the [^18^F]FMISO uptake was significantly higher in 12-day-A2058-COX-2KO.

### 3.5. COX-2 Knockout Reduces the Plating Efficiency but Does Not Effect Chemosensitivity and Radiosensitivity

To quantify the viability of the cells after treatment with different concentrations of the selective COX-2 inhibitors, celecoxib and rofecoxib, we determined the activity of succinate dehydrogenase by the conversion of tetrazoliumbromide. Reduced viability was observed only at high concentration of celecoxib (50 µM), i.e., 40% viability for A2058 and about 20% for A2058-COX-2KO (Figure 6a). Lower concentrations of celecoxib (12.5 and 25 µM) and all concentrations of rofecoxib (12.5, 25 and 50 µM) rarely affect the viability of both cell lines, which is between 70–90%. For the further tests, non-toxic concentrations of 1 µM and 10 µM of both inhibitors were used. Next, we performed a colony formation assay. In our experiments, a colony is defined to consist of at least 50 cells. This method was used to determine cell reproductive death after treatment with COX-2 inhibitors, and further in combination with ionizing radiation.

In the colony formation assay, significantly fewer colonies were formed by A2058-COX-2KO cells compared to A2058 (50 ± 15 vs. 130 ± 30 colonies/well or plating efficiency of 0.06 vs. 0.02, respectively) (Figure 6b). To determine whether celecoxib or rofecoxib had effects in a COX-2-dependent or COX-2-independent manner, the cells were treated with 1 µM or 10 µM celecoxib or rofecoxib, respectively. As shown in Figure 6b, celecoxib had no effects on the colony formation in A2058 cells, whereas low concentration of celecoxib significantly enhanced the colony formation in A2058-COX-2KO. Rofecoxib showed the tendency also to enhance the survival of the colonies in both A2058 and A2058-COX-2KO cells (Figure 6b). Of note, the plating efficiency of A2058-COX-2KO after celecoxib or rofecoxib application was in the same range as the plating efficiency of untreated native A2058 cells.

The relative plating efficiency of both cell lines was compared after different doses of X-ray radiation. Although COX-2 knockout cells had generally a significant reduced plating efficiency of about 40%, radiation effects were almost identical for both cell lines (Figure 6c). In another experiment, COX-2-related effects on clonogenic survival were investigated after treatment with both X-ray radiation and further incubation with celecoxib or rofecoxib. In both cell lines, irradiation inhibited the colony formation in a dose-dependent manner, and high doses of 6 Gy or 10 Gy showed nearly complete growth inhibition. In A2058 cells, irradiation in combination with celecoxib or rofecoxib at 1 µM or 10 µM showed that only 10 µM celecoxib had a tendency to reduce the clonogenic survival (Figure 6d). Low concentration of celecoxib and rofecoxib at 1 µM and 10 µM resulted in a trend towards increased survival. These effects were clearer in A2058-COX2-KO cells (Figure 6e). Here, a significant increase in survival after irradiation (2, 4 and 6 Gy) was observed under rofecoxib (1 µM and 10 µM) treatment. For celecoxib, an additionally low concentration at 1 µM in combination with 2 Gy caused a significant increase in survival.

Together, COX-2 knockout reduces the colony formation in A2058 melanoma cells of about 40%, but radiation effects of X-rays are identical for both cell lines. Incubation with selective COX-2 inhibitors revealed an increased plating efficiency in cells with COX-2 knockout, which seems to be protective for A2058 melanoma cells. Treatment of A2058-COX-2KO cells in combination with X-ray and specific inhibitors further supported these protective effects. In addition, radiosensitivity and chemosensitivity were not influenced and seem to be independent of COX-2 expression. Here, celecoxib and rofecoxib did not exert COX-2 specific effects in both A2058 and A2058-COX-2KO melanoma cells, and additional COX-2-independent pathways have to be considered.

### 3.6. Altered COX-2 Downstream Signaling in COX-2 Knockout Cells

Based on the characterization of the obtained COX-2 knockout model, differences in the signaling cascades influencing the regulatory pathways associated with the synthesis/activation of COX-2 should be confirmed. In order to identify potential downstream signaling targets of COX-2, we compared the two cell lines A2058 and A2058-COX-2KO using a “Cancer Phospho Signaling Antibody Array” (269 targets) in a pilot experiment. Using the normalized data, the fold change between A2058 and A2058-COX-2KO was determined and the results are highlighted as red (increase in expression, ratio ≥ 2) or green (decrease in expression, ratio ≤ 0.5) (Appendix A). The results for the “Cancer Phospho Signaling Antibody Array” are shown in Figure 7. In the A2058-COX-2KO cells, many signaling proteins were significantly downregulated compared to the untreated A2058 cell line. The different expression analysis of the proteins is displayed in Figure 7a. Most of the signaling proteins for the PI3 kinase/Akt, the MAPK and Jak were markedly decreased in A2058-COX-2KO (Figure 7b).

## 4. Discussion

The COX-2 enzyme plays a complex role in melanoma progression and chemoresistance by stimulating cell proliferation, cell invasion, inducing vessel formation and enhancing metastasis and immunosuppression [40]. Moreover, COX-2 influences patients’ survival and represents a marker for metastatic development [8,12]. High COX-2 expression and activity is not only found in cancer cells, but also in cancer-associated fibroblasts and macrophages [7] and seems to be an effective gene target in the TME due to its function as a key mediator of inflammation. Therefore, a blockade of COX-2 could be an effective tool to prevent or treat cancer.

Several reports have demonstrated that silencing or full deletion of COX-2 significantly inhibited melanoma cell proliferation, cell invasion and motility in vitro [10,12,28,29,30,31]. Ercolano et al. previously generated a COX-2 knockout in mouse B16F10 melanoma cells by the CRISPR/Cas9 technique and observed marked effects on invasiveness and motility in vitro and tumor growth in vivo [29]. For our work, we decided to use the human melanoma A2058 cell line. The main reason was to avoid the major problem of the adhesion and growth factor profiles of B16F10, which are unlike those of its human melanoma counterparts. In B16F10, the enzymes used for invasion into tissues, the ability of cells to overpower the immune system, the antiapoptotic mechanisms and many other cancer cell hallmarks do not reflect the human disease [41]. Taking into account the heterogeneity of the available melanoma cell lines, which is demonstrated, among other things, by the fact that melanoma cell can show different secretion pathways for the same protein among themselves, our observation is more of a basic finding on a frequently used specific melanoma cell line than a generalizable finding with immediate specific clinical relevance in melanoma [42].

Our results show reduced subcutaneous tumor growth in vivo with A2058-COX-2 knockout tumor xenografts compared to the corresponding A2058 tumor xenograft, which is consistent with the findings by Ercolano et al. [29]. Moreover, A2058-COX-2KO cells exhibited a significantly reduced invasion, colony and spheroid formation potential in cell monolayers as well as in three-dimensional cell models in vitro. In order to further characterize the effects of COX-2 knockdown in vitro, metabolic parameters (hypoxia), sensitivity to treatment with COXIBs and/or irradiation and the influence on signaling pathways were investigated in monolayer culture and additionally in multicellular tumor spheroids in addition to the growth rate.

COX-2 upregulation within the TME in response to hypoxia via stabilization of hypoxia-inducible factor-1 (HIF-1) and nuclear factor kappa-light-chain-enhancer of activated B cells (NF-κB) has already been detected [4]. In fact, hypoxic regions of the tumor positively correlate with increased COX-2 expression, accumulation of M2-like polarized macrophages and other immunosuppressive cells [43]. A hypoxia-induced increase in COX-2 protein expression in A2058 cells by simultaneous upregulation of hypoxic markers, for instance, CA-IX and HIF-1α, has also been demonstrated by our group [38]. As expected, hypoxia in the A2058-COX-2KO cells did not turn-on COX-2 protein expression after 24, 48 and 72 h (data not shown). In our work, we studied both oxygen-restricted adherent cell monolayer cultures as an extrinsic model and 3D-tumor spheroids as an intrinsic hypoxia model. The development of hypoxic conditions in the extrinsic hypoxia model was confirmed in the A2058 cells by measurements of the hypoxia-affine radiotracer [^18^F]FMISO. However, hypoxia effects on COX-2 knockout cells were less distinct. In order to further clarify this, both cell lines were used to generate spheroids. In the present investigation, three different time points with different spheroid diameters were used as an intrinsic hypoxia model. For spheroids, it has been reported that the switch between normoxic and intrinsic hypoxic conditions occurs in the diameter range around 300 and 600 µm [36,44], which also seems to explain the observed variability in radiotracer uptake [45]. The A2058-COX-2KO spheroids grew faster than A2058 spheroids and showed a marked difference in morphology. A2058 cells spontaneously form a single rounded spheroid with distinct hypoxic areas, further substantiated by immunohistochemical staining of pimonidazole. In contrast, the A2058-COX-2KO aggregate into loose clusters of irregular shape and uptake of [^18^F]FMISO was quite low over time. Again, the COX-2 knockout spheroids pimonidazole staining was reduced at day 19, indicating spreading necrotic areas. Together, the data indicated a connection between COX-2 knockout, reduced [^18^F]FMISO uptake and increased formation of necrosis. Furthermore, the reduced uptake of [^18^F]FDG in A2058 and COX-2 knockout spheroids on day 19 is a sign of altered metabolic activity, which can be explained in spheroids particularly by the enlargement of the hypoxic core zone or necrotic areas during growth or an extended culture period [36].

Moreover, knockout of COX-2 showed inhibitory effects on the colony formation capacity of about 40% in vitro. In this context, the influence of irradiation and treatment with selective COX-2 inhibitors was investigated. As recently published, irradiation of A2058 cells resulted in an upregulation of COX-2, depending on the irradiation dose and the time after irradiation [38]. In the COX-2KO cells, these upregulation effects were not observed (data not shown). In general, A2058 melanoma cells have an overall low plating efficiency compared to other cancer cell lines. Interestingly, irradiation did not change the relative plating efficiency between these two cell lines, and COX-2 seems not to increase radiosensitivity. Furthermore, pro-apoptotic and anti-proliferative effects of celecoxib and rofecoxib had no effect on the plating efficiency. Only high concentrations of celecoxib and rofecoxib affected the cell viability, but this did not seem to be a result of COX-2 dependent actions. Interestingly, this phenomenon was also observed in another human melanoma cell line (Mel-Juso), with rarely detectable COX-2 expression [38], and in human neuroblastoma cell line SH-SY5Y [21]. For celecoxib, several alternative mechanisms have been described, including inhibition of protein-dependent kinase (PDK) [46,47], carbonic anhydrase [27] and 5-lipoxygenase [48]. In contrast, rofecoxib, a more potent selective COX-2 inhibitor, has only marginal PDK inhibitory activity [22]. COX-2-independent mechanisms have also been identified in human prostate cancer cells that express only COX-1 [49]. The authors observed anticarcinogenic effects with low concentrations of celecoxib and an increased treatment period, whereas rofecoxib had no effect on cell growth over the same concentration range. These findings are confirmed by our own data in A2058-COX-2 knockout cells. For celecoxib and rofecoxib, a COX-2-specific effect could not be observed, instead the concentration used in our experiments promoted the plating efficiency.

Radiosensitivity enhancement by treatment with selective COX-2 inhibitors (celecoxib and rofecoxib) could only be detected by high dose single X-ray irradiation (6 and 10 Gy). In contrast, treatment with low irradiation doses and low concentrations of rofecoxib (1 and 10 µM) significantly increased the cell survival in the clonogenic assay. These effects were similar for low concentrations of celecoxib, and more pronounced in the COX-2 knockout cells. Shin et al. observed radiosensitization as a result of treatment with COX-2 selective inhibitors via a COX-2 protein-dependent mechanism in human lung and colon adenocarcinoma cells. The same effects have been observed for human breast carcinoma cells with different COX-2 expression levels using high doses of celecoxib (30 and 50 µM) [26]. While it is supposed that COX-2 increases radiosensitivity, the mechanisms are not yet fully understood. It is accepted that COX-2 inhibitors sensitize cancers to ionizing radiation through both COX-2-dependent and independent mechanisms. Our data indicated that both radiosensitivity and chemosensitivity are not influenced in A2058 melanoma cells and seem to be independent of COX-2 expression. Furthermore, celecoxib and rofecoxib did not act by COX-2 dependent effects, and COX-2-independent pathways have to be considered. The so called off-target-effects of celecoxib have already been described [50]. Here, binding affinity to proteins other than the COX-2 enzyme with relation to tumor metastasis and angiogenesis are pointed out.

Finally, we compared the expression level of signaling pathway-related proteins in A2058 vs. A2058-COX-2KO cells using a “Cancer Phospho Signaling Antibody Array”. Here, we verified significantly altered downstream signaling pathways associated with COX-2 in cellular and molecular mechanisms of cancer metastasis [7]. COX-2 knockout significantly changed the protein expression characteristics of melanoma cells and affected several signaling pathways closely related to tumorigenesis. The knockout of COX-2 mainly resulted in the downregulation of various related proteins, which accounts for the diverse function. COX-2 is known to activate the PI3K/AKT and MAPK signaling pathways, leading to cancer cell survival and inflammation [3,10,51]. Here, most proteins related to this signaling pathway are downregulated in A2058-COX-2 knockout cells. More importantly, COX-2 is a mediator of angiogenesis and tumor growth through upregulation of vascular endothelial growth factor (VEGF) and of epidermal growth factor receptor (EGFR), respectively [9,52]. Both factors are significantly downregulated in A2058-COX-2 knockout cells. Proteins for p53- and NFкB-signaling, cell cycle progression and apoptosis ensure cell survival are downregulated [53].

As recently published, effective treatment of metastatic melanoma by inhibition of the MAPK and PI3K pathways strongly encourage further novel treatment approaches [17]. Zhou et al. demonstrated that a combination therapy of PKCζ and a COX-2 inhibitor suppresses B16-F10 melanoma cell migration, invasion and adhesion, and blocked lung metastasis of intravenously injected B16-F10 cells into C57BL/6 mice [54].

## 5. Conclusions

In conclusion, our results confirm the role of COX-2 in melanoma progression. Certainly, knockout of COX-2 indicates anti-tumorigenic properties in melanoma cells regarding colony formation, migration and invasion, and finally, tumor growth in vivo. However, due to the complexity of signaling pathways and multitasking roles of COX-2, the data in this study imply a suppression of COX-2, in combination with the inhibition of related pathways, is suggested to be a promising therapeutic approach for melanoma treatment.

## Figures and Tables

**Figure 1 cells-11-00749-f001:**
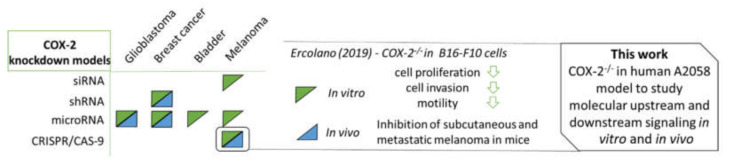
Schematic representation of approaches to generate COX-2 knockdown/out models in vitro (green triangle) and in vivo (blue triangle) [10,12,29,30,31,33]; selected results from Ercolano et al. [29] and the aim of this work.

**Figure 2 cells-11-00749-f002:**
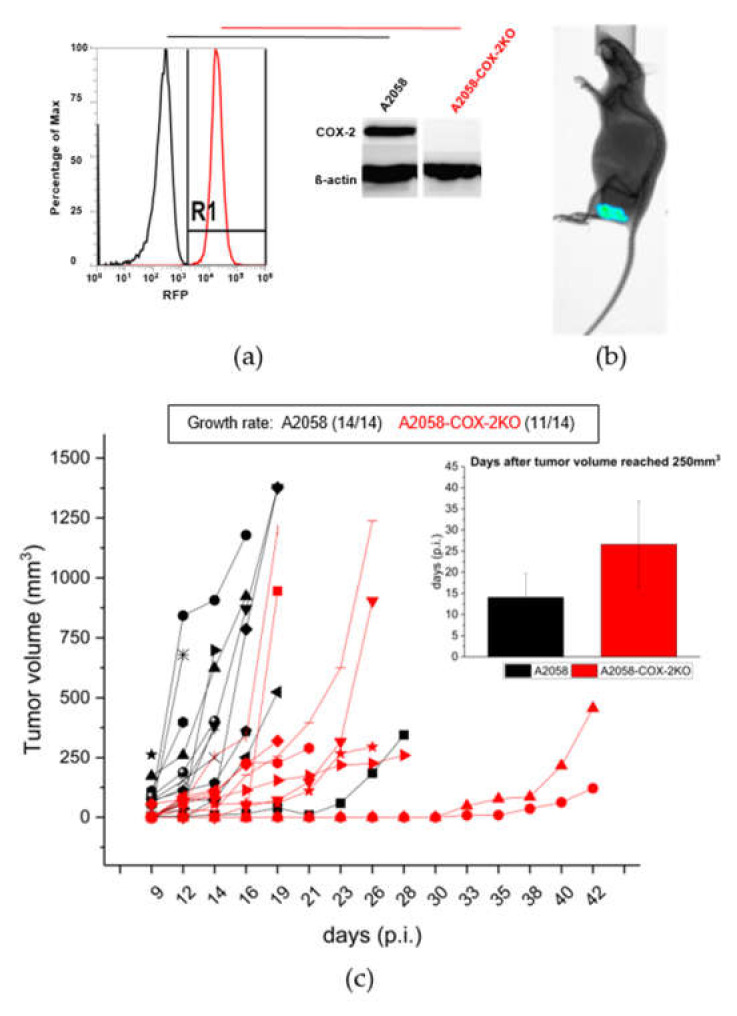
Confirmation of COX-2 knockout with CRISPR/Cas9 in vitro and in vivo. (**a**) Knockdown of COX-2 using CRISPR/Cas9 in A2058 melanoma cells was confirmed by RFP-FACS (red) and Western blot analysis compared to the control cells (black). (**b**) Representative bioluminescence image showing A2058-COX-2KO cells transplanted subcutaneously in NMRI-nude mice. (**c**) Tumor growth rate and progression analysis after the subcutaneous injection of A2058 (black, n = 14) and A2058-COX-2KO (red, n = 11) cells in NMRI-nude mice. Values are presented as mean ± SD.

**Figure 3 cells-11-00749-f003:**
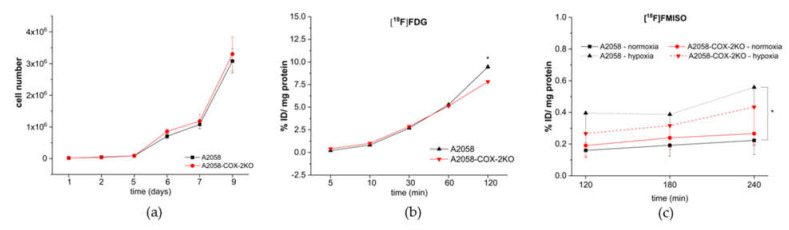
Characterization and comparison of monolayer A2058 (black) and A2058-COX-2KO cells (red). (**a**) Analysis of cellular growth of A2058 and A2058-COX-2KO over 9 days (n = 3). (**b**) Uptake of [^18^F]FDG as a parameter of glucose metabolism over 120 min (n = 6). (**c**) [^18^F]FMISO uptake under normoxic and hypoxic conditions in monolayer cell cultures over a time period of 240 min (n = 6). Data are shown as mean ± SEM of three independent experiments, * *p* < 0.05. In (**c**), only negative error bars are shown. Note that the SEM bars in (**b**) do not exceed the size of the symbols.

**Figure 4 cells-11-00749-f004:**
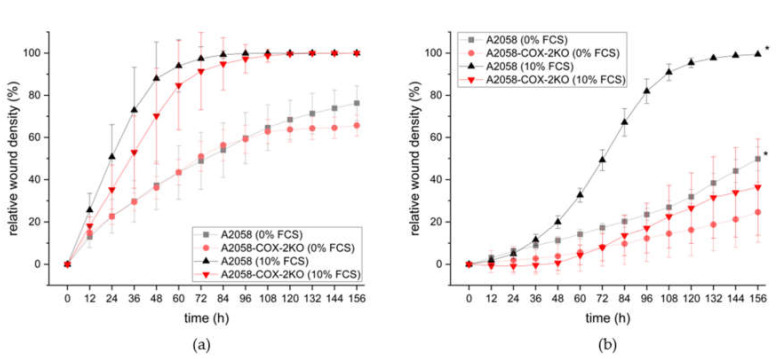
Knockdown of COX-2 inhibits cell motility of monolayer A2058 (black) and A2058-COX-2KO cells (red). (**a**) The migration potential and (**b**) invasion properties through Matrigel were analyzed in a scratch wound healing assay over 7 days. Cells were imaged every 12 h in proximity of the wounds under phase contrast (10× objective) using the IncuCyte^®^ Live-Cell Monitoring System. Data are shown as mean ± SEM of two independent experiments.

**Figure 5 cells-11-00749-f005:**
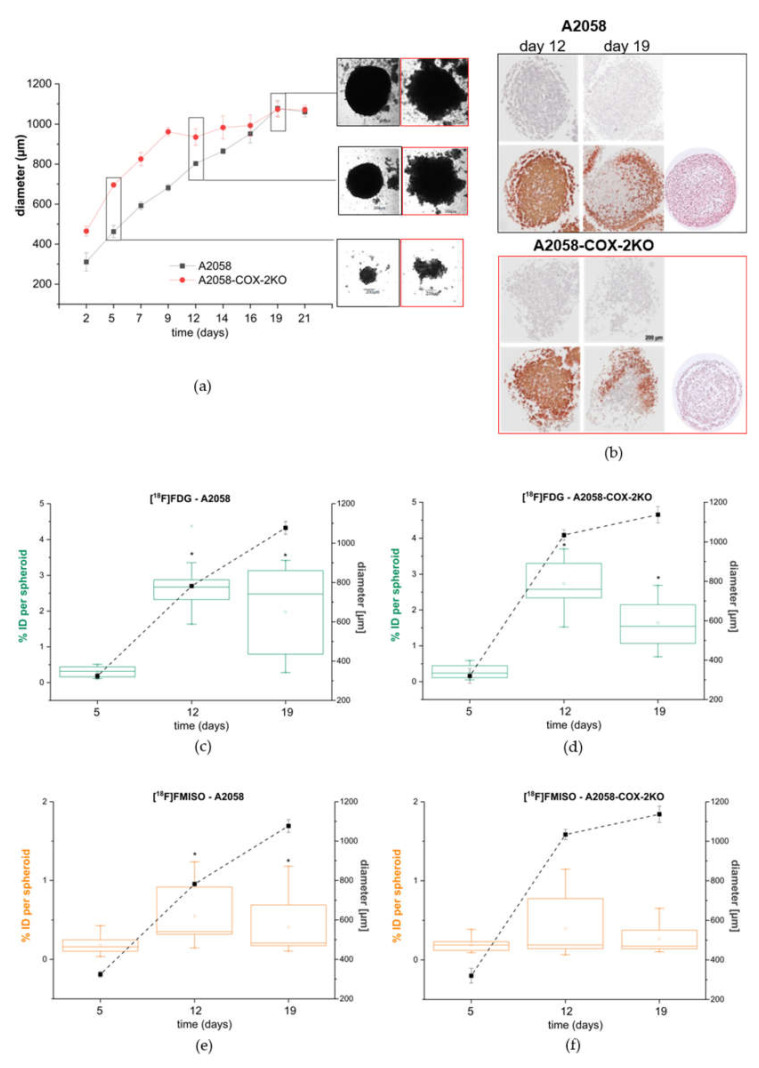
Characterization and comparison of A2058 and A2058-COX-2KO spheroids. (**a**) Growth curves of A2058 and A2058-COX-2KO over 21 days (n = 12) and representative images of spheroids after day 5, 12 and 19. (**b**) Pimonidazole immunostaining of 12 days and 19 days for A2058 and A2058-COX-2KO spheroids. Control section without the primary antibody is shown above the pimonidazole images. HE staining of 19-day spheroids are displayed on the right. (**c**, **d**) Uptake of [^18^F]FDG in (**c**) A2058 and in (**d**) A2058-COX-2KO after 5, 12 and 19 days (n = 6). The green boxes show the uptake of the radiotracer; the black dotted line shows the spheroid diameter. (**e**, **f**) Uptake of [^18^F]FMISO in (**e**) A2058 and in (**f**) A2058-COX-2KO after 5, 12 and 19 days (n = 6). The orange boxes show the uptake of the radiotracer; the black dotted line shows the spheroids diameter. Data are shown as mean ± SEM of three independent experiments, * *p* < 0.05.

**Figure 6 cells-11-00749-f006:**
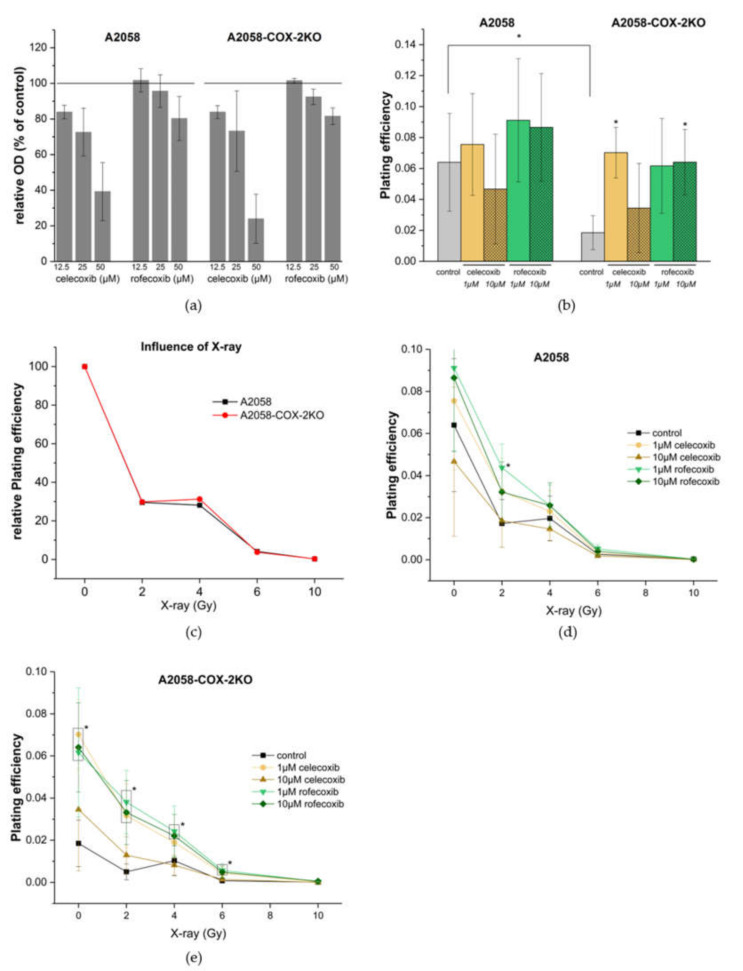
Influence of celecoxib or rofecoxib and X-ray on cell viability and clonogenicity. (**a**) Cell viability after treatment with different concentrations of celecoxib or rofecoxib was determined by MTT assay and is expressed as optical density (OD) values. (**b**) Plating efficiency of A2058 and A2058-COX-2KO after treatment with celecoxib and rofecoxib using a clonogenic assay (n = 8). (**c**) Relative plating efficiency after X-ray. Here, plating efficiency of untreated A2058 and A2058-COX-2KO cells were set to 100% and the relative plating efficiency after irradiation with 2, 4, 6 and 10 Gy was determined. (**d**,**e**) Plating efficiency of A2058 and A2058-COX-2KO after X-ray and treatment with celecoxib and rofecoxib using a clonogenic assay (n = 8). Data are shown as mean ± SEM of three independent experiments, * *p* < 0.05.

**Figure 7 cells-11-00749-f007:**
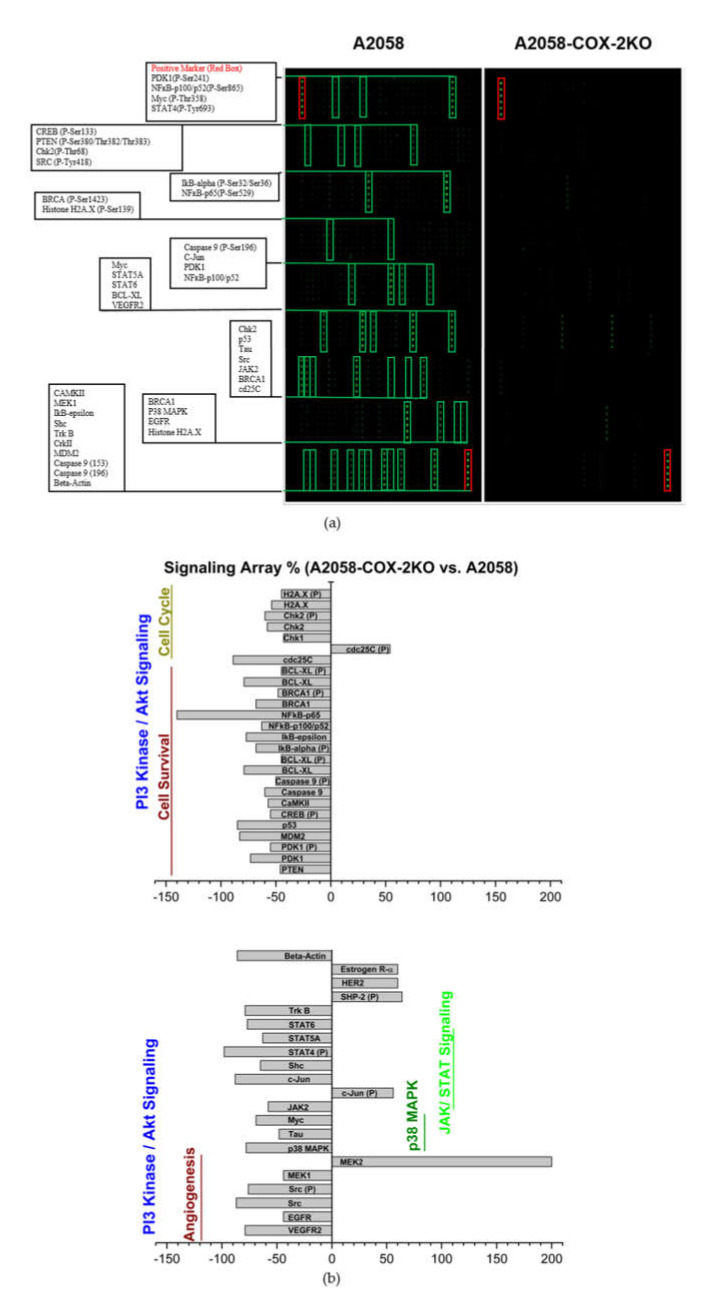
One sample analysis of *“Cancer Phospho Signaling Antibody Array”*. (**a**) The array images are shown for both arrays: A2058 (left) and A2058-COX-2KO (right), and the most prominently changed proteins are marked in green rectangles with six replicates for each antibody. Positive references are marked in red rectangles. (**b**) The most changed downstream signaling molecules are displayed on a vertical-bar chart, and displayed as a percentage of the median value of the average signal intensity for each antibody compared to A2058 control cells.

## Data Availability

The data presented in this study is contained within the article and Appendix A.

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
