# Peer review of "CRISPR/Cas9 Mediated Knockout of Cyclooxygenase-2 Gene Inhibits Invasiveness in A2058 Melanoma Cells"

_cells, 2022, doi:10.3390/cells11040749_

Round 1

Reviewer 1 Report

A study by Haase-Kohn et al investigates a functional role of COX-2 in melanoma cells using CRISPR/Cas9-mediated knockout. The manuscript is well written and demonstrates technically valid data.

The limitations of the study are:

  1. the use of a single melanoma cell line;
  2. the reference to not current literature

Author Response

Response to Reviewer 1 Comments

A study by Haase-Kohn et al investigates a functional role of COX-2 in melanoma cells using CRISPR/Cas9-mediated knockout. The manuscript is well written and demonstrates technically valid data.

The limitations of the study are:

Point 1: the use of a single melanoma cell line;

Response 1: Thank you for this critical comment. We had already performed comparable experiments with two other A2058-COX-2KO clones. We found no differences between these clones in terms of monolayer cell growth (Figure S3) and subcutaneous tumor growth (Figure S4). We also perform experiments of another tumor cell line (human glioblastoma). Our results to date on this non-melanoma cell line confirm our experiments reported here. These studies are ongoing and unpublished.

However, our focus was malignant melanoma. From our point of view, another (model) cell line, even more so with a different ontogenetic origin, could yield different results and therefore complicate the conclusions on melanoma as presented here.

On the other hand, we have revised the manuscript as such that our results were obtained with a specific melanoma cell line. Considering the enormous heterogeneity and molecular variability of the various available melanoma cell lines and of the clinical picture itself, our findings can therefore not be directly translated to the clinic application for malignant melanoma patients.

Point 2: the reference to not current literature

Response 2: Thanks for the kindly remind. We included the literature into the current list.

Citation 32 (page 2 - lane 73 and page 20 - lane 680-682)

Front Vet Sci. 2021 Aug 26;8:633170. doi: 10.3389/fvets.2021.633170. eCollection 2021.

COX-2 Silencing in Canine Malignant Melanoma Inhibits Malignant Behaviour

Reviewer 2 Report

The manuscript by Haase-Kohn and colleagues reports their findings regarding the impacts of COX-2 knockout in A2058 melanoma cell line in vitro and in xenograft model.

Although the results are somewhat interesting, the finding is in line with other such reports and thus lacks novelty. There are also major flaws in experimental design and execution as reported. I have some major concerns regarding the experiments:

-Is the KO cell line that the authors created for COX-2 a monoclonal or polyclonal population. It is only mentioned that puro resistant and RFP positive cells were collected “single-cell isolation”.

-If a polyclonal population was used, there should still be some COX-2 expression (for eg in western blots). If a monoclonal population was used, a control cell which has gone through the same selection and/or sorting process needs to be used as a control in the all the experiments reported. As has been reported, A2058 wild type cells can not serve as proper controls.

-The authors need to provide Sanger sequencing data for the exon of COX-2 that was targeted for KO.

- Figure 2 says 14/14 and 11/14 for the two groups, not sure what that means. Please mention. Western blot images are cut way too tight. Provide full gel image.

-Figure 4: Provide representative images for scratch assays.

- In figure 3b, add time point(s) after 120 mins for FDG. i.e. Is it more significant after 2 hours?

-fig 5 a error bar missing. In fact, there isn’t much information to gain from Figure 5 other than Fig 5a.

- Lines 81-85 needs to be re-written.

-Only minor spell checks and sentence rewriting required.

Author Response

Response to Reviewer 2 Comments

The manuscript by Haase-Kohn et al. described the role of isoenzyme cyclooxygenase-2 (COX-2) in mediating aggressive feature of A2058 melanoma cell line. Overall the results are interesting, even if the impact of COX-2 in tumors, including also melanoma, was previously addressed. The knockout of COX-2 using CRISPR/Cas9 method, compared to old methods to silence COX-2 previously exploited, is a plus of the paper. However, I suggest to the authors different experiments to improve the conclusions.

Major points:

Point 1: All the results were obtained using only 1 cell line A2058, and only, in my opinion, 1 KO clone of A2058. Usually is better to validate results also in a second KO clone and I would like to see at least the in vivo experiments and the invasion assays using a second cell line. This is very important to sustain the conclusions.

Response 1: Thank you for this critical comment. We had already performed comparable experiments with two other A2058-COX-2KO clones. We found no differences between these clones in terms of monolayer cell growth (Figure S3) and subcutaneous tumor growth (Figure S4). We also perform experiments of another tumor cell line (human glioblastoma). Our results to date on this non-melanoma cell line confirm our experiments reported here. These studies are ongoing and unpublished.

However, our focus was malignant melanoma. From our point of view, another (model) cell line, even more so with a different ontogenetic origin, could yield different results and therefore complicate the conclusions on melanoma as presented here.

On the other hand, we have revised the manuscript as such that our results were obtained with a specific melanoma cell line. Considering the enormous heterogeneity and molecular variability of the various available melanoma cell lines and of the clinical picture itself, our findings can therefore not be directly translated to the clinic application for malignant melanoma patients.

Point 2: It would be better to show and include “data not shown” that are important

Response 2: We agree with the reviewer. We added the data concerning the PGE2 levels to the supplementary data (Figure S2) and replaces “data not shown” by Figure S2 (lane 246) Furthermore, we added data on COX-2 expression after hypoxia and after X-ray exposure (Figure S8). On the behalf of the editor we would add this figure additional to the supplemental section.

Point 3: Paragraph 3.2: The discrepancy between the decreased tumor growth in vivo in KO cells and the comparable rate of proliferation in vitro needs more discussion. Regarding the metabolic features of these cells I suggest to implement the metabolic measurements analyzing lactate secretion and activity of some metabolic enzymes involved in glycolysis. It would be great if the authors could perform analysis of metabolic fluxes using Seahorse technology, if available. Hypoxia could be also evaluated in tumors in vivo using pimomidazole staining as the authors used for spheroids.

Response 3: We agree with the reviewer’s concern. In our opinion, the difference between the observed in vitro and in vivo results is most likely caused by the surrounding microenvironment. Under standard in vitro conditions, it can be assumed that cells off both lines are surrounded by optimal nutrient availability, resulting in a similar growth rate. In vivo conditions present a tumour microenvironment, where insufficient nutrients or nutrient gradients exists resulting in an advantage for the wild-type cells. We have added an appropriate comment in the results section (page 7 - line 267-273). Instead of lactate secretion or pimonidazole (ex vivo) uptaken we employed radiopharmacological assays. [18F]FDG and [18F]FMISO are standard clinical methods that allow to accurately quantify metabolic changes such as glucose uptake/energy status and hypoxia. Unfortunately, the Seahorse technology is not available at our institute.

Point 4: Paragraph 3.3: Include images of wound-healing in the Figure and add an assay using classical invasion in matrigel using transwell.

Response 4: The Incucyte system acquired images from all time points during wound healing. These images now have been included in the supplementary information (Figure S5 and S6) with the references in the text page 8 lane 301 for S5, and in lane 307 for S6. However, the data presented in these images is not part of the growth curves (Fig. 4), which is the reason for only showing them in the supplementary. Moreover, the ‘classical’ invasion test with Matrigel using transwell slides was also performed with the Incucyte system. Here we measured at three different time points (0, 24, 48 and 72 hours). The results are consistent with the results of our invasion and migration experiments. We have included the figures in the supplementary (Figure S7).

Point 5: Paragraph 3.4: Include 2D colony-forming assay to evaluate basal growth. The increased growth of spheroids of KO cells is in contrast with a decreased proliferation in vivo. I understood that these clones are different in morphology compared to parental cells but I do not understand why in 3D assays KO cells growth faster….this point merits a deeper discussion and explanation.

Response 5: The KO-cell based spheroids are characterized more as loose clusters. The typical formation of solid and compact spheroids was not observed here. While the diameter is increased, the total number of cells is not significantly different from the A2058 wild-type cells. We have revised the manuscripts results accordingly (page 9 - lane 332-334).

Point 6: Paragraph 3.5: In my opinion it would be better to move this paragraph as last result. Data from these results clearly demonstrated that there are off-targets of COX-2 inhibitors and that in this cellular model COX-2 inhibition did not increase sensitivity to radio/chemo treatments. These results are weak…thinking to the possible clinical use of COX-2 inhibitors. It would be interesting try to combine COX-2 inhibitors and BRAF/MEK inhibitors for example commonly used in melanoma.

Response 6: We agree mostly with the reviewer’s conclusion. In the manuscript, we describe COX-2-independent effects of celecoxib/rofecoxib. However, we did not consider in more detail the potential off-target mechanisms described so far. No major differences between cell lines were observed in this regard, however, a COX-2-independent radiosensitizing effect of celecoxib/rofecoxib was found. We have revised the discussion and included two papers that have described such off-target aspects in COXIB application in melanoma and in other pathologies (references 49: page 17 – 547-549 and page 21 - lane 723-724).

Point 7: Paragraph 3.6: Add analysis of the main signaling pathways deregulated in KO cells using western blot.

Response 7: We thank the reviewers for this comment. However, we believe that such experiment are beyond the scope of this manuscript. The array data presented here mainly servers as a proof of principle, showing that COX-2 knockout affects a number of signaling pathways. The large number of targets investigated allows a first estimation of which signaling pathways are affected and which are not. However, and here we fully agree with the reviewer's concern, this is only a preliminary result that needs to be substantiated by further experiments. Consequently, the manuscript was amended to make this point clear.

Reviewer 3 Report

The manuscript by Haase-Kohn et al. described the role of isoenzyme cyclooxygenase-2 (COX-2) in mediating aggressive feature of A2058 melanoma cell line. Overall the results are interesting, even if the impact of COX-2 in tumors, including also melanoma, was previously addressed. The knockout of COX-2 using CRISPR/Cas9 method, compared to old methods to silence COX-2 previously exploited, is a plus of the paper. However, I suggest to the authors different experiments to improve the conclusions.

Major points:

  1. All the results were obtained using only 1 cell line A2058, and only, in my opinion, 1 KO clone of A2058. Usually is better to validate results also in a second KO clone and I would like to see at least the in vivo experiments and the invasion assays using a second cell line. This is very important to sustain the conclusions.
  2. It would be better to show and include “data not shown” that are important
  3. Paragraph 3.2: The discrepancy between the decreased tumor growth in vivo in KO cells and the comparable rate of proliferation in vitro needs more discussion. Regarding the metabolic features of these cells I suggest to implement the metabolic measurements analyzing lactate secretion and activity of some metabolic enzymes involved in glycolysis. It would be great if the authors could perform analysis of metabolic fluxes using Seahorse technology, if available. Hypoxia could be also evaluated in tumors in vivo using pimomidazole staining as the authors used for spheroids.
  4. Paragraph 3.3: Include images of wound-healing in the Figure and add an assay using classical invasion in matrigel using transwell.
  5. Paragraph 3.4: Include 2D colony-forming assay to evaluate basal growth. The increased growth of spheroids of KO cells is in contrast with a decreased proliferation in vivo. I understood that these clones are different in morphology compared to parental cells but I do not understand why in 3D assays KO cells growth faster….this point merits a deeper discussion and explanation.
  6. Paragraph 3.5: In my opinion it would be better to move this paragraph as last result. Data from these results clearly demonstrated that there are off-targets of COX-2 inhibitors and that in this cellular model COX-2 inhibition did not increase sensitivity to radio/chemo treatments. These results are weak…thinking to the possible clinical use of COX-2 inhibitors. It would be interesting try to combine COX-2 inhibitors and BRAF/MEK inhibitors for example commonly used in melanoma.
  7. Paragraph 3.6: Add analysis of the main signaling pathways deregulated in KO cells using western blot.

Minor points:

     1. Check for typos and grammar errors in the text.

     2. In the Discussion it would be better to discuss some mixed results   obtained (for example proliferation in vivo/in vitro, spheroids growth rate)

Author Response

Response to Reviewer 3 Comments

The manuscript by Haase-Kohn and colleagues reports their findings regarding the impacts of COX-2 knockout in A2058 melanoma cell line in vitro and in xenograft model.

Although the results are somewhat interesting, the finding is in line with other such reports and thus lacks novelty. There are also major flaws in experimental design and execution as reported. I have some major concerns regarding the experiments:

Point 1: Is the KO cell line that the authors created for COX-2 a monoclonal or polyclonal population. It is only mentioned that puro resistant and RFP positive cells were collected “single-cell isolation”.

Response 1: The knockout cell line used in our experiments are collected from single cells, and are mentioned as a monoclonal population. We added this comment in the sentence on page 3 lane 115-116.

RFP positive and puromycin resistant cells were selected by single-cell collection (monoclonal population), and named A2058-COX-2KO.

 Most experiments were performed with 3 different monoclonal clones to verify the results.

Point 2: If a polyclonal population was used, there should still be some COX-2 expression (for eg in western blots). If a monoclonal population was used, a control cell which has gone through the same selection and/or sorting process needs to be used as a control in the all the experiments reported. As has been reported, A2058 wild type cells can not serve as proper controls.

Point 3: The authors need to provide Sanger sequencing data for the exon of COX-2 that was targeted for KO.

Response 3: For optimal reaction efficiency with the Cox-2 CRISPR/Cas9 knockout plasmid we did the transfection together with target-specific HDR plasmid which enables precise gene editing at the DSB site. During repair, the Cox-2 HDR plasmid incorporates a Red Fluorescent protein (RFP) to visually confirm transfection and a puromycin resistance gene for selection of cells containing a successful CRISPR/Cas9 double-strand break. RFP positive and puromycin resistant cells were selected by single-cell collection (monoclonal population) to confirm complete allelic knockouts. We finally confirmed and characterized the Cox-2-knockout by gene expression monitoring via fluorescent microscopy and Cox-2 Western blotting.

Point 4: Figure 2 says 14/14 and 11/14 for the two groups, not sure what that means. Please mention. Western blot images are cut way too tight. Provide full gel image.

Response 4: 14/14 means: that in the A2058 control group 14 of the subcutaneously injected animals developed a tumor. Despite, the COX-2 knockout cells only 11 out of 14 animals a tumor has grown.

We added a sentence on page 6 lane 255-257: In animals injected with A2058 wild-type cells, tumors grew in 14 of 14 animals, and in animals injected with A2058-COX-2KO only 11 animals of 14 developed tumors.

Full Western Blot images are provided in the supplementals S1.

Point 5: Figure 4: Provide representative images for scratch assays.

Response 5: Representative images for the scratch assay are provided in the supplementals S5 and S6.

Point 6: In figure 3b, add time point(s) after 120 mins for FDG. i.e. Is it more significant after 2 hours?

One could expect that in this experiment a longer observation time could have shown greater difference in FDG accumulation (the activity of [18F] is measured decay-corrected at a physical half-life of about 110 minutes for [18F]) between the two cell lines at later time points. However, considering the radiopharmacological behavior of FDG standardized radiotracer uptake measurements in cells usually cover a period of 1 to 2 hours.

Point 6: fig 5 a error bar missing. In fact, there isn’t much information to gain from Figure 5 other than Fig 5a.

Response 6: We included the error bars in figure 5a.

Point 7: Lines 81-85 needs to be re-written.

Response 7: Page 2 - line 81-85 has been re-written

Here, subcutaneous tumor growth in vivo as well as migration, invasion and colony formation, growth rate and metabolic parameters (hypoxia) were investigated. Furthermore, sensitivity to treatment with COXIBs and/or irradiation as well as the influence on signaling pathways in monolayer cultures and/or multicellular tumor spheroids was measured in vitro.

Round 2

Reviewer 2 Report

The authors have addressed most of the comments.

Reviewer 3 Report

The revised version of the paper is now acceptable for publication.